# Modification of Natural and Synthetic Zeolites for CO_2_ Capture: Unrevealing the Role of the Compensation Cations

**DOI:** 10.3390/ma18102403

**Published:** 2025-05-21

**Authors:** Norberto J. Abreu, Andrés F. Jaramillo, Daniel F. A. Becker-Garcés, Christian Antileo, Rebeca Martínez-Retureta, Jimmy A. Martínez-Ruano, Jaime Ñanculeo, Matías M. Pérez, Mara Cea

**Affiliations:** 1Departamento de Ingeniería Química, Facultad de Ingeniería y Ciencias, Universidad de La Frontera, 01145 Francisco Salazar, Temuco 4780000, Chile; danibecker@udec.cl (D.F.A.B.-G.); christian.antileo@ufrontera.cl (C.A.); jimmy.martinez@ufrontera.cl (J.A.M.-R.); m.perez15@ufromail.cl (M.M.P.); mara.cea@ufrontera.cl (M.C.); 2Centro de Manejo de Residuos y Bioenergía, BIOREN, Universidad de La Frontera, 01145 Francisco Salazar, Temuco 4780000, Chile; j.nanculeo01@ufromail.cl; 3Department of Mechanical Engineering, Universidad de La Frontera, 01145 Francisco Salazar, Temuco 4780000, Chile; 4Departamento de Ingeniería Mecánica, Universidad de Córdoba, Cr 6 #76-103, Montería 230002, Colombia; 5Departamento de Ingeniería Química, Universidad de Concepción, Concepción 4070386, Chile; 6Departamento de Ciencias Ambientales, Facultad de Recursos Naturales, Universidad Católica de Temuco, Rudecindo Ortega 02950, Temuco 4780000, Chile; rebeca.martinez@uct.cl; 7Programa de Doctorado en Ciencias de la Ingeniería Mención Bioprocesos, Universidad de La Frontera, 01145 Francisco Salazar, Temuco 4780000, Chile; 8Laboratorio Químico, División Chuquicamata, SGS Chile, Calama 1390000, Chile

**Keywords:** carbon capture and storage, CO_2_ adsorption, ion exchange, isotherm modelling, zeolite

## Abstract

The development of highly effective natural-based adsorbents to face the increasing rates of CO_2_ production and their delivery to the atmosphere are a big concern nowadays. For such purposes, synthetic and natural zeolites were modified via an ion exchange procedure to enhance the CO_2_ uptake. Samples were characterized by SEM, EDS, TGA and nitrogen adsorption at 77 K, showing the correct incorporation of the new metals; in addition, the CO_2_ adsorption isotherms were determined using a gas analyser. During the first stage, the role of the compensation cations for CO_2_ adsorption was assessed by modifying a pure ZSM-5 synthetic zeolite with different metal precursors present in salt solutions via an ion exchange procedure. Then, five samples were studied; the samples modified with bivalent cation precursors (Zn^2+^ and Cu^2+^) presented a higher adsorption uptake than those modified with a monovalent cation (Na^+^ and K^+^). Specifically, the substitution of the compensation cations for Cu^2+^ increased the CO_2_ capture uptake without affecting the surface properties of the zeolite. The results depict the prevalence of π-cation interactions enhanced by the field gradient induced by divalent cations and their lower ionic radii, if compared to monovalent ones. Subsequently, a natural zeolite was modified considering the best results of the previous phase. This Surface Response Methodology was implemented considering 11 samples by varying the concentration of the copper precursor and the time of the ion exchange procedure. A quantitative quadratic model to predict the adsorption uptake with an R^2^ of 0.92 was obtained. The results depicted the optimal conditions to modify the used natural zeolite for CO_2_ capture. The modification procedure implemented increased the CO_2_ adsorption capacity of the natural zeolite more than 20%, reaching an adsorption capacity of 75.8 mg CO_2_/g zeolite.

## 1. Introduction

The increasing demands of human activities and industrial development necessitate intensive energy consumption, primarily through the combustion of fossil fuels. However, these processes lead to significant environmental challenges, particularly due to the high production rates of carbon dioxide (CO_2_) and other greenhouse gases. CO_2_, which accounts for approximately 65% of the global radiative forcing caused by long-lived greenhouse gases, acts as a significant absorber and emitter of infrared radiation [1]. Over the past 50 years, atmospheric CO_2_ concentrations have risen by more than 100 ppm, contributing to global warming and ocean acidification [2]. CO_2_ alone accounts for approximately 74% of global anthropogenic greenhouse gas (GHG) emissions and is the major contributor to climate-related impacts [3]. According to the Intergovernmental Panel on Climate Change (IPCC) [4] and recent economic analyses [5], the global economic losses resulting from climate change, predominantly driven by GHGs such as CO_2_, could reach 11–14% of GDP under a scenario of 2 °C warming by 2050, meaning potential losses higher than USD 11 trillion.

Carbon capture and storage (CCS) technologies have emerged as effective methods for mitigating CO_2_ emissions. Projections indicate that CCS technologies could sequester up to 240 billion tonnes of CO_2_ by 2050 [2]. Among the existing CCS, the traditional approaches, such as amine-based absorption, could achieve over 90% efficiency [6,7]. Nevertheless, operational challenges, including high energy consumption, solvent degradation, and equipment corrosion, underscore the necessity for improved technologies [8]. Alternative technologies, particularly solid adsorbents, are emerging as promising solutions. These systems offer lower energy consumption, reduced corrosion risks, and adsorption efficiencies of up to 95% [9,10]. Notably, replacing amine-based absorption systems with solid adsorbents can reduce the energy required for solvent regeneration by approximately 40% [7].

Among the available solid adsorbents, zeolites and activated carbons stand out due to their exceptional properties [11]. Specifically, zeolites are crystalline aluminosilicates with a highly ordered pore structure, ranging from 0.3 to 1 nm in diameter, that allow for selective adsorption and permeation based on molecular size and polarity [12]. Their thermal stability, resistance to acids and bases, and modifiable surface properties make them versatile for several applications, such as gas stream purification and natural gas upgrading [13]. Additionally, adsorbents based on zeolites can be reactivated by chemical and thermal processes, allowing them to be used several times [14]. Moreover, the high mechanical stability of zeolites, attributed to their advanced nanomechanical properties, makes these materials suitable for large-scale processes. In this sense, zeolite can be used as a component for concrete for building, using functional materials for industrial applications [15]. For practical, large-scale application, the production costs should allow the adsorption to be competitive with other CO_2_ separation processes.

The unique structure of zeolites comprises a three-dimensional tetrahedral framework of SiO_4_ and AlO_4_ units. The substitution of Si with Al induces a negative charge, which is balanced by exchangeable cations, often alkali cations, within the porous network. These compensating cations maintain the structure’s electro-neutrality and confer specific sorption characteristics, including ion exchange and selective adsorption [13,16]. The functionalization of zeolite surfaces enables the incorporation of new chemical and physical properties into these materials. In particular, supporting transition and noble metals on zeolites is a widely employed functionalization strategy for adsorbent and catalyst synthesis, as it yields materials with high surface areas and enhanced chemical activity [17].

Carbon dioxide adsorption in zeolites is predominantly influenced by the interaction between CO_2_ molecules and the electric field generated by compensating cations [18]. In the last few years, several studies concerning CO_2_ remotion have been conducted using zeolites, both synthetic zeolites, such as Y [19,20], 13X [21,22], LTA [23], SSZ13 [24], and natural zeolites such as clinoptilolite [10,11,25,26] and mordenite [25,27,28]. Different experiments have been performed to determine the main structural and experimental factors influencing the carbon dioxide adsorption onto zeolites. In this sense, the influence of parameters, such as the temperature [29,30,31], pressure [30,32,33], Si/Al ratio [29,34,35], compensation cation [32,36,37], metal loading [35,38,39], among other properties, has been assessed. Insights drawn from research reviews [17,37] depict that CO_2_ adsorption is principally governed by the inclusion of exchangeable cations within the cavities of zeolites, which induce the electric field interacting with the adsorbate molecule. Research also suggests that basic compensating cations enhance the electrostatic field on the zeolite surface, improving CO_2_ adsorption [6]. Interestingly, acidic zeolites have also demonstrated promising results, underscoring the need for further investigation into the role of cations in enhancing the adsorption uptake.

Zeolites offer a promising pathway toward sustainable industrial practices. Their availability, particularly of natural zeolites, reduces production costs, enhancing their applicability [40]. Even when previous studies have reported insights into the role of the compensation cations of zeolites and other parameters in CO_2_ adsorption, few reports have checked the optimal conditions to modify natural zeolites to obtain a cost-effective CO_2_ adsorbent. This work investigates this potential through a two-stage approach. Initially, the effect of modifying synthetic zeolites by ion exchange with various compensating cations on surface physicochemical properties and CO_2_ adsorption capacity was evaluated. Building on these findings, the second stage focused on modifying natural Chilean zeolites with salts of specific compensating cations to improve CO_2_ adsorption, aiming to develop efficient and cost-effective adsorbent materials. This research contributes to advancing carbon capture technologies and promoting green industrial processes as the CO_2_ trapped in these materials can be further valorised to interesting energetic resources, such as Methanol and Dimethyl ether, via a catalytic process. Thus, this work aligns with Sustainable Development Goal (SDG) 7 “affordable and clean energy” and Goal 13 “climate action”, specifically by integrating climate change-related measures by diminishing the GHGs emitted by the industrial process and the conventional energy production systems, among others [41].

## 2. Materials and Methods

### 2.1. Materials and Reagents

Synthetic ZSM-5 zeolite provided by TOSOH Corporation (Grove City, OH, USA) was employed. Specifically, the 840NHA variant containing NH_4_⁺ ions as compensating cations was utilised. In addition, the natural zeolite was sourced from a natural extraction site in Maule, located in Chile’s VII Region, and provided by the company Zeolita del Maule (Maule, Chile). According to the technical datasheet, this zeolite consists of a mixture of Clinoptilolite and Mordenite.

For the chemical modification of the zeolite samples, the following salts were used as precursors: copper (II) nitrate trihydrate [Cu(NO_3_)_2_·3H_2_O], zinc nitrate hexahydrate [Zn(NO_3_)_2_·6H_2_O], sodium nitrate [NaNO_3_], and potassium nitrate [KNO_3_]. These salts were supplied, respectively, by Merck S.A. (Darmstadt, Germany), Sigma Aldrich (Burlington, MA, USA), Winkler (Santiago, Chile), and Merck S.A. (Darmstadt, Germany), all with analytical-grade purity.

The use of a synthetic and mono-cationic zeolite of known composition was aimed at investigating the role of the compensation cations and their interaction with the CO_2_ molecule. This effect is particularly difficult to investigate in poly-cationic zeolites, such as natural zeolites, where the ion exchange equilibrium is more complex as well as the final composition after modification [42]. Therefore, once the cations that enhanced CO_2_ capture were identified, the incorporation of this specific cation into the synthetic zeolite was studied to optimize the surface.

### 2.2. Physicochemical Modification of Zeolites

The parent zeolites were modified by chemical and thermal processes, considering an ion exchange procedure and a thermal outgassing in an inert atmosphere. Before the modifications, the zeolites were washed several times with deionized water and dried in an oven (WGLL-BE, FAITHFUL, Cangzhou, China) at 363 K for 24 h.

#### 2.2.1. Chemical Modifications by Ion Exchange Process

The precursor solutions used for modifying the parent zeolites were prepared using a nominal mass percentage, calculated as grams of the target cation ($w_{cation}$) per gram of the final modified zeolite ($w_{cation}$ + $\;w_{zeolite}$). In this sense, the weight of the metal cation within the precursor salt molecule was determined considering the ratio of their molar mass. Consequently, the theoretical percentage of the metal cation within the modified zeolite was calculated using Equation (1).(1)$$Cation\;\%=\frac{{w\;}_{cation\;}\left(g\right)}{w_{cation\;}\;\left(g\right)+w_{zeolite}\;\left(g\right)}\times 100$$

In this work, Na⁺, K⁺, Cu^2^⁺, and Zn^2^⁺ were selected as compensating cations based on their specific physicochemical characteristics, such as ionic radius, charge density, and their ability to promote interactions with CO_2_ molecules [13,37,43]. In addition, the selection considered cost-effective salt precursors among the possible cations. For the initial phase of this study, a nominal percentage of 4% was selected.

Afterward, the zeolite was added to the precursor solutions at a ratio of 10:1 (cm^3^ of solution per gram of modified zeolite), and the samples were placed in a thermostatically controlled water bath at 363 K using an EV400H rotary evaporator (Labtex, Jinan, China). The ion exchange process was conducted under constant stirring at 120 rpm for 4 h. Subsequently, the samples were washed in deionized water and oven dried (Labtex, China) at 363 K for 24 h and stored in a vacuum desiccator for future use. Values of metal loading, exchange temperature and the procedure time were determined based on prior investigations involving natural and ZSM-5 zeolites modified with copper cations [44,45]. The modification parameter selection also considered information from other researchers [43,46,47].

For a better understanding, samples are labelled as indicated in Table 1.

It is noteworthy that, during the second phase of this study, the precursor solutions were prepared at varying concentrations, and the ion exchange process was conducted for different durations.

#### 2.2.2. Thermal Outgassing of Chemically Modified Zeolites

To remove residues from the ion exchange process and other volatile impurities from the zeolite surface, the pre-modified samples were placed in a tubular reactor supported by a quartz frit, under a continuous argon flow of 300 cm^3^/min. The setup was subjected to a gradual heating process in an SZGL-1200C furnace (SIOMM, Shanghai, China) at a constant temperature increase rate of 5 K per minute, reaching 623 K, where it was held for one hour. The cooling phase was subsequently performed while maintaining the argon flow.

### 2.3. Characterization of the Adsorbent Materials

#### 2.3.1. Porosimetry Measurements

The textural characteristics of the parent zeolites and the prepared adsorbents were evaluated through nitrogen adsorption–desorption isotherms at 77 K using a Nova 800 porosimeter (Anton Paar, Graz, Austria). The surface area was calculated using the Brunauer–Emmett–Teller (BET) method, while the BJH (Barrett–Joyner–Halenda) method for pore size distribution, H-K (Horvath–Kawazoe) method for micropore volume and t-plot method for micropore surface area estimation were applied. Prior to analysis, the samples were degassed at 623 K for 2 h to eliminate potential interference with pore accessibility.

#### 2.3.2. Scanning Electron Microscopy

The morphology of the samples was examined through scanning electron microscopy (SEM) with an SU-3500 microscope (Hitachi, Tokyo, Japan), operated under high vacuum (30 Pa) at an accelerating voltage of 10.0 kV. Images were captured at magnifications ranging from 5 μm to 100 μm. Energy-dispersive spectroscopy (EDS) was employed alongside SEM to confirm the incorporation of copper nanoparticles within the zeolite framework.

#### 2.3.3. Thermogravimetric Measurements

Thermogravimetric Analysis (TGA) was conducted using an STA 6000 analyser (PerkinElmer, Shelton, CT, USA) to evaluate the thermal stability of the samples and quantify the mass loss associated with the degassing process. During the assay, the samples were heated from room temperature to 873 K at a constant heating rate of 15 K per minute. Furthermore, mass variations were recorded in parallel with heat flow measurements using a Differential Scanning Calorimeter (DSC) coupled to the STA 6000 analyser.

### 2.4. Determination of the Maximum CO_2_ Adsorption Capacity

#### 2.4.1. Experimental CO_2_ Isotherms

The CO_2_ adsorption isotherms were obtained using a NOVA 800 gas analyser (Anton Paar, Graz, Austria). For this analysis, 0.06 g of zeolite were pre-activated through an outgassing procedure under vacuum. The temperature was increased at a rate of 10 K/min from room temperature to 573 K, held for 3 h, and then cooled to 298 K. Subsequently, a flow of pure CO_2_ was applied through the cell containing the samples at relative pressures (p/p_0_) ranging from 0.001 to 1. The temperature remained constant at 298 K throughout the experiment. The precision for gas adsorption measurements is given as ±1% of reading and reproducibility of 2% as reported by the manufacturer.

#### 2.4.2. Isotherm Modelling

The experimental data of CO_2_ adsorption isotherms were analysed using the linearised Langmuir and Freundlich models to determine the adsorption capacity and the affinity of CO_2_ for the original and modified zeolites. The Langmuir isotherm was considered appropriate due to the presumed monolayer adsorption and surface homogeneity of the synthetic zeolite, while the Freundlich model accounted for the potential heterogeneity and multilayer adsorption effects, particularly at higher concentrations, mainly because the study involved the use of natural zeolites [48]. Table 2 displays the isotherm models used.

### 2.5. Determination of the Optimised Modification Conditions for Natural Zeolites

Response Surface Methodology (RSM) was applied employing a Central Composite Design (CCD) to optimise the conditions that enhance the adsorption capacity of the natural zeolite (NZ_0). This experimental design incorporated a factorial approach with two levels, three central points and four-star points, yielding a total of 11 samples. The analysis was conducted using MODDE PRO-13 software. The software randomized the order of the experiments to reduce bias in the execution of the measurements [49]. The statistical analysis will be performed considering the correlation coefficient (R^2^) and cross validation correlation coefficient (Q^2^) referring the adjustment of the experimental data to those predicted by the model.

The independent variables considered were the theoretical copper concentration to be incorporated onto the zeolite surface (2% to 8%), calculated using Equation (1), and the duration of the ion exchange process (4 to 12 h). Table 3 presents the experimental design, which comprised eleven samples with theoretical copper concentrations ranging from 0.34% to 11% and ion exchange durations from 2.34 to 13.7 h. It is worth noting that, to generate the response surface, the CCD included modification conditions that extended beyond the initially defined variable limits.

## 3. Results and Discussion

### 3.1. On the Influence of the Compensating Cation on the CO_2_ Adsorption in Synthetic Zeolite

In the first experimental stage, a pure synthetic zeolite was used to assess how the incorporation of new compensation cations affects the CO_2_ adsorption behaviour. In this sense a commercial ZSM-5 zeolite synthetized with NH_4_^+^ as compensating cation was used. Then, ion exchange procedures in identical conditions but with different metal precursors were performed.

#### 3.1.1. Physical, Chemical and Surface Characterization of Raw and Modified Synthetic Zeolites

Table 4 displays information about the physical-chemical composition and surface properties of the synthetic zeolite and the samples generated by modifying their compensation cations. As expected, the parent zeolite (SZ) presents only Si, Al and O (Nitrogen and Hydrogen cannot be detected by the EDS apparatus) whose, purity allowed to verify the incorporation of the desired metallic cations via ion exchange and their influence in the zeolite properties. In addition, the incorporation of the new compensation cations was verified in all the samples at concentrations ranging from 1.17% (SZ_Na) to 2.03% (SZ_K) without affecting the Si/Al ratio.

The nitrogen adsorption assays yielded interesting results. A slight decrease in the surface area was observed across samples exchanged with divalent cation precursors, while, for the sample prepared using KNO_3_ and NaNO_3_ as the precursor, the decrease in the total surface was slightly higher. Moreover, in the cases using Cu(NO_3_)_2_ and Zn(NO_3_)_2_ as cation precursors, the microporous surface remained similar and a slight decrease in the mesoporous area occurred in both samples. The samples modified with monovalent cations presented decreases in both the microporous and mesoporous surfaces. However, based on the lower decrease of the microporous and mesoporous surfaces for all the samples, it can be inferred that the modification procedure was successfully implemented without affecting the main structure of the zeolite. Regarding the pore volume, the sample presented similar results with a lower effect on the samples modified with Cu^2+^ and Zn^2+^ precursors. Finally, it is noting worth that the average pore size is similar for all the studied samples.

SEM images and the Elemental Mapping performed using EDS (Figure 1) confirmed the successful incorporation of the metals in the zeolite. At first inspection, SEM image of SZ (Figure 1A) looks similar to the modified samples (Figure 1C,E,G,I) suggesting that the external surface was not affected. EDX spectra is displayed in the Appendix A. The Peaks referring to carbon element correspond to the material used to deposit the samples during the sample preparation for imaging.

Besides, Figure 1D,F,H,J displays the metal dispersion of the metal elements in the modified samples. As it can be observed, all the samples present a uniform distribution of the metals added by ion exchange, specifically, the formation of metal clusters or zones without the used cation was not observed. It is interesting to note that, in the sample modified with KNO_3_ (Figure 1F), it can be appreciated a higher particle density consistent with the higher concentration of K element obtained by EDS analysis.

#### 3.1.2. On the Determination of the Temperature for the Outgassing Procedure Using Thermogravimetric Information

The results of the Thermogravimetric Analysis are displayed in Figure 2. For all samples, the first peaks of weight loss occur around 363 K. This result can be attributed to the desorption of water from the zeolite surface [50]. Consequently, aligned with the observations of Sun et al. [51], water seems to be mostly desorbed during the initial degassing stage, in a temperature range from 300 K to 523 K. Moreover, further changes that occurred at temperatures higher than 523 K can be attributed to the desorption of volatile compounds such as the compound formed by the anion that accompanies the metallic precursor deposited as a compensation cation on the zeolite surface. The desorption at these higher temperatures could also be attributed to more tightly bound water molecules located in the micropores as well as the dihydroxylation by the remotion of some OH groups from the zeolite framework [52].

The parent zeolite (Figure 2A) exhibits a second desorption peak around 573 K to 773 K, characteristic of the NH_3_ desorption from strongly acidic sites [53], allowing for the zeolite provided of Brönsted acidic sites by creating new Si-O-H terminals [45]. On the other hand, the second desorption peak occurred at lower temperatures on the modified samples (Figure 2B–D), with almost negligible weight loss after 623 K, suggesting that NH_4_^+^ cations were correctly replaced during the ion exchange process. Thus, the released substances are likely the nitrate used during the modification process. In this sense, it was checked that the minimum temperature required to activate the modified zeolites was approximately 623 K, since around 80% of the species were desorbed from the surface at this temperature.

Likewise, the heat flow graphs confirm that the mass loss processes are of an exothermic nature [52,54,55], in agreement with those discussed above. Specifically, an endothermic transition is observed at lower temperatures, like the weight loss attributed to water desorption. In addition, at higher temperatures, broader peaks can be observed, also suggesting endothermic transitions, centred at temperatures similar to those at which the mass loss peaks were reported. Finally, glass transition signals were not detected, either in the pristine or modified samples, which is also a typical behaviour of highly crystalline zeolites such as the ZSM-5 zeolite used. It is worth noting that the heat flow graphs are reported with an inverted axis (endothermic up) for a better visualization.

#### 3.1.3. Assessing the Role of the Zeolite Compensation Cations for CO_2_ Adsorption

As expected, the CO_2_ adsorption isotherms at 298 K suggest that the cation exchange procedure affects the interaction between the target molecule and the adsorbent. Figure 3 indicates that modifying synthetic zeolite with divalent cations, within the studied concentration range, could enhance the interaction between the CO_2_ molecule and the zeolite surface. According to the shape of the adsorption curves, a typical type 1 isotherm is formed, suggesting the formation of an adsorbate monolayer. In the low-concentration range, where pressures are below 10 kPa, the adsorption capacity increases almost linearly with the CO_2_ concentration. As the CO_2_ concentration rises further, the surface appears to reach saturation, resulting in a minimal increase in absorbed volume, which ultimately levels off. This behaviour is characteristic of the adsorption process that primarily occurs through chemical interactions in materials with a regular porous structure, such as the synthetic zeolites utilized in this study [56].

As suggested by the adsorption curve shape, the experimental data closely align with the Langmuir model for the raw ZSM-5 zeolite and the modified samples, demonstrated by a coefficient of determination (R^2^) greater than 0.99. It can, therefore, be stated that localized chemical adsorption mainly takes place, meaning that the CO_2_ molecules are retained as a monolayer on the zeolite surface [57]. In contrast, the limited alignment of the data with the Freundlich model, with R^2^ values ranging from 0.93 to 0.977 in the most accurately fitting datasets, suggests that the formation of multiple adsorbate layers on the zeolite surface could be discarded [58].

Interesting results were obtained from the Langmuir model. Although the CO_2_ uptake by zeolites modified with monovalent and divalent cations presents clear differences at the experimental concentration range, the maximum CO_2_ uptake ($Q_{max}$) predicted by the Langmuir (Table 5) model remains similar for all samples (105.4–105.9 mg/g). However, the $Q_{max}$ value obtained is calculated at the curve asymptote, which may not necessarily correspond to the adsorbate concentration range relevant in applying the adsorbent materials.

The fact that approximately 70% of CO_2_ emissions originate from anthropogenic sources represents a global issue. Kumar and Nagendra et al. (2015) investigated CO_2_ emissions in commercial, industrial, and urban sectors of Chennai, India, reporting values of 467 ± 33.45 ppm, 464 ± 31.68 ppm, and 448 ± 33.45 ppm, respectively [59]. A similar case has been documented in the Hsinchu Industrial Park, Taiwan, where a daily average of nearly 600 ppm was recorded [60]. Another significant source of CO_2_ emissions is vehicular transport, with an annual average concentration of 533 ± 105 ppm in Delhi, India, highlighting the urgent need for cost-effective CO_2_ capture solutions [61]. In this concentration range, the zeolite samples modified with copper and zinc precursors present higher adsorption uptakes than the other samples applicable to the CO_2_ uptake from contaminant sources.

The results obtained are concordant with previous works that reported high adsorption capacities for molecules with a high quadrupolar moment when using zeolites modified with divalent cations, as opposed to those modified with monovalent cations, particularly in the case of copper-modified zeolites [45,62].

According to this, the behaviour of the adsorption can be explained by two main aspects. On the one hand, the field gradient within the zeolite and quadrupole interaction seems to dominate the adsorption of CO_2_, as the adsorbate possesses a strong quadrupole moment but no dipole [63]. In this sense, Cu^2+^ and Zn^2+^ as divalent cations have twice the amount of charge than Na^+^ and K^+^ with monovalent cations. On the other hand, previous studies have reported that the π-cation interactions present an inverse proportionality with the ionic radii [45,63]. In this sense, higher adsorption was obtained using the Cu^2+^- and Zn^2+^-modified samples, with ionic radii of 73 and 74 pm, respectively, lower than Na^+^ and K^+^ with ionic radii of 102 pm and 138 pm, respectively. Accordingly, molecular dynamics simulations have also suggested that the interactions among the CO_2_ (or other quadrupolar molecules) and zeolite can be improved by exchanging zeolites with cations with lower ionic radii [64,65,66]. This phenomenon can be explained by a shorter distance between the centre of mass of the CO_2_ molecule and the cation, increasing the Van der Waals interactions [64,66].

### 3.2. On the Development of a CO_2_ Adsorbent Material Based on Natural Zeolite

Considering the results obtained in the previous sections, the lower price of natural zeolites compared to synthetics and their availability in Chile, in the second phase of this study, natural Chilean zeolites were modified with copper cation precursors.

#### 3.2.1. Physicochemical and Surface Characterization of Raw and Modified Natural Zeolite Samples

Five representative samples of natural zeolites were selected from the RSM experimental design (Table 3) to study the influence of chemical modification on the physical, chemical, and surface properties. For a better understanding, samples were ordered considering the amount of copper salts used in the precursor solutions. In this sense, the selected samples were NZ (0%), NZ_5 (0.34%), NZ_3 (2%), NZ_8 (6%), NZ_4 (10%).

Table 6 displays information about the physicochemical composition and surface properties of the natural zeolite and the samples generated. As has been reported in previous investigations using natural Chilean zeolite, its chemical composition differs from the synthetic zeolite with several compensation cations, such as Na, Mg, K, Ca and Fe. It is important to note that in all the modified samples, copper was detected; however, the mass percentage of this metal was lower than the calculated concentration. Nevertheless, this is normal behaviour for ion exchange procedures because the zeolite has a limited quantity of compensating cations and also because some of the compensating cations preset in the natural zeolite possess a high affinity to the surface, becoming difficult to remove [67]. In this sense, in all the modified samples, the mass percentage of the initial cations diminished, while the copper concentration reached 2.76% in the sample with higher copper loading.

Regarding the surface properties, all the modified natural zeolites presented higher surface areas and pore volumes than the raw natural zeolite. This result was also expected because the chemical and physical modification can clean the zeolite surface from impurities and previously adsorbed compounds in their natural reservoirs. Similar to what occurred with synthetic zeolites, the variation in the surface areas was low, and there were no evident variations in the external surface, as can be seen in Figure 4, suggesting that the incorporation of the metal within the zeolite does not block the pore network.

#### 3.2.2. CO_2_ Adsorption Isotherms onto Raw and Modified Natural Zeolites

Table 7 presents a summary of the obtained parameters for Langmuir and Freundlich mathematical models applied to the CO_2_ adsorption onto the natural zeolite (Z0) and selected samples, of natural zeolites modified with copper (Z5, Z3, Z8 and Z4). The experimental data adjusted to the Langmuir model for the selected samples are shown in Figure 5. Here, it can be observed that the parameter $Q_{max}$ of the Langmuir model, which indicates the maximum adsorption capacity of CO_2_ in the zeolites, increases in most of the modified samples compared to the unmodified sample (Z0). The Langmuir model parameters for all the samples are presented in Appendix A.

Specifically, the $Q_{max}$ values for samples Z3, Z4 and Z8 are higher than those of Z0, indicating an improvement in the adsorption capacity due to the modification of the zeolites. However, sample Z5 presented a slightly lower adsorption capacity than Z0, which could be due to the low concentration of copper added, which might not be sufficient to improve this capacity significantly.

As expected, the experimental data fit the Langmuir model well. Similar to what occurred with the synthetic zeolites, in the explored concentrations, the CO_2_ uptake and the maximum adsorption capacity increase, indicated by the Langmuir model, which increases from 61.72 mg/g in the natural unmodified zeolite (NZ) to 73.5 mg/g in the sample NZ_4. This result can be attributed to the higher amounts of the copper precursor, if compared to those used to modify synthetic zeolite. Additionally, contrary to the results obtained using synthetic zeolites, the CO_2_ adsorption isotherms using the raw and modified natural zeolite samples also fit the Freundlich model. In this sense, the Freundlich model parameter K_f_, which is directly related to the adsorption capacity in heterogeneous systems, presented results reaffirming that copper modifications in natural zeolites increase their CO_2_ adsorption capacity. Furthermore, the values of the parameter n in all samples suggest the occurrence of physisorption processes [68], in which the active sites of natural zeolites (raw and modified) may have the capacity to adsorb one or more CO_2_ molecules, with a non-parallel orientation, thus indicating a multi-molecular adsorption process for each sample [69].

#### 3.2.3. Finding the Optimized Conditions to Modify Natural Zeolites for CO_2_ Capture: SRM Results

Table 8 presents the results of the maximum CO_2_ adsorption capacity for each sample, obtained directly from the analysis of experimental isotherms according to the Langmuir model as well as the model parameters.

Figure 6A shows the Surface Response as a bidimensional representation of the polynomial response when analysing how ion exchange time and the amount of copper in the precursor solution affect the CO_2_ adsorption capacity of natural zeolite. The interaction between a high ion exchange time and an increase in copper concentration is indicated in the reddish area of the graph as the optimal conditions with the highest CO_2_ capture efficiency. In addition, the model parameters presented in Figure 6B allow for exploring the relationships between the copper concentration in the precursor solution (Cu) and the ion exchange time (t) with the CO_2_ adsorption capacity (Q_e_).

Similar to the preliminary analysis, the square parameter that indicates the interactions between both variables presents a positive value, so it can be assumed that the best modification conditions are obtained by combining high values in both variables, at least in the experimental range. It is well known that, when carrying out ion exchange processes with very saturated precursor solutions, it can lead to the precipitation of these metals as oxides on the surface of the support, blocking the pores, decreasing the surface area and finally affecting the removal capacity.

It is worth noting that the statistical parameters related to the obtained model presented show a high level of fit, considering that it is a quadratic model (R^2^ = 0.92). This can be corroborated in Appendix A (see Appendix A), where it can be observed that the values obtained by the model are similar to the values obtained experimentally.

According to the data obtained, the optimized conditions to modify the natural zeolite for the remotion of CO_2_ are those considering a precursor solution with a theoretical copper concentration of 9.06% (w_Cu_/w_Zeolite+Cu_) and performing the modification process by ion exchange by 10.9 h, predicting a maximum adsorption uptake of 76.3 mg/g, increasing the adsorption uptake of the natural zeolite at 23.6%. The optimized sample was also developed in this investigation. The maximum adsorption uptake of CO_2_ obtained by fitting the CO_2_ adsorption data to the Langmuir model was 75.8 mg/g, which closely aligns with the predicted value, supporting the validity of the model within the studied range. To facilitate a better understanding of the model, a three-dimensional Response Surface Plot is included in the Appendix A.

Finally, a comparison between the removal capacity of the natural zeolites and the synthetic zeolite studied is necessary. In this sense, although the removal capacity of the synthetic zeolite is indeed more significant than the CO_2_ adsorption capacity of the natural zeolite, the surface area of the synthetic zeolite almost triplicates the values of the natural one. Table 9 shows a weighted comparison for the surface area.

It is well known that several aspects different from the Surface Area could affect the adsorption uptake, such as the pore size, pore volume, the polarity and the nature and distribution of the active sites [70], among another parameters. In this case, it is evident that the amount of CO_2_ adsorbed per m^2^ of surface is more significant in the case of natural zeolites, and such results can be attributed to the presence of diverse monovalent and divalent compensation cations present in the natural zeolite and the possibility of multilayer adsorption, as suggested by the good fit to the Freundlich model. Furthermore, the existence of bigger zeolite channels in the natural zeolite, specifically the site pocket of mordenite, could alter cation localization. It has been suggested that the allocation of cations in the existing larger pores leaves channels of zeolites free for absorbate transport, which is beneficial for adsorption [71].

A possible modification strategy could involve physicochemical treatments to the natural zeolite to increase its surface area before the ion exchange procedure. The zeolites developed using this methodology could also be investigated for the remotion of other hazardous compounds from gaseous streams and/or their valorization to valuable chemical products.

## 4. Conclusions

Ion exchange methodology has proven to be an efficient technique to modify the compensating cations of synthetic and natural zeolites, without affecting their structure or morphological characteristics. This methodology promotes a new charge balance in the original zeolite structure, allowing for the incorporation of more adequate cations for specific processes such as CO_2_ adsorption. Using precursor solutions with salts of divalent cations such as Cu(NO_3_)_2_ and Zn(NO_3_)_2_ improves the CO_2_ adsorption capacity of zeolites by increasing the field gradient on their surface. In this way, cations that present more significant interaction with the pi bond electrons in the CO_2_ molecule would be incorporated. The adsorption uptake was also favoured by the lower ionic radius in the case of Cu^2+^ and Zn^2+^ compared to monovalent cations such as Na^+^, K^+^ and NH_4_^+^. At high CO_2_ concentrations, the adsorption uptake of all the samples reaches 105–106 mg of CO_2_ per zeolite gram.

The Response Surface Methodology allowed us to determine the influence of the concentration of the copper precursor solutions and the modification times on the CO_2_ adsorption capacity of natural zeolites, showing that, in the range of variables studied, the increase in copper contraction in the precursor solution and the ion exchange times favour the process; however, it is observed that, above certain values in both variables, the performance of the material is affected. Finally, the optimization of these parameters allowed for obtaining a catalyst with a removal capacity of 75 mg_CO2_/g_ads_, which increased the maximum adsorption capacity of the natural zeolite by 23.6%. The modified zeolites developed in this investigation are promising and cost-effective adsorbent materials suitable for use in filters for CO_2_ capture. Furthermore, future studies could explore their application in the removal of other hazardous compounds from gaseous streams and/or the valorization of these compounds into valuable chemical products.

## Figures and Tables

**Figure 1 materials-18-02403-f001:**
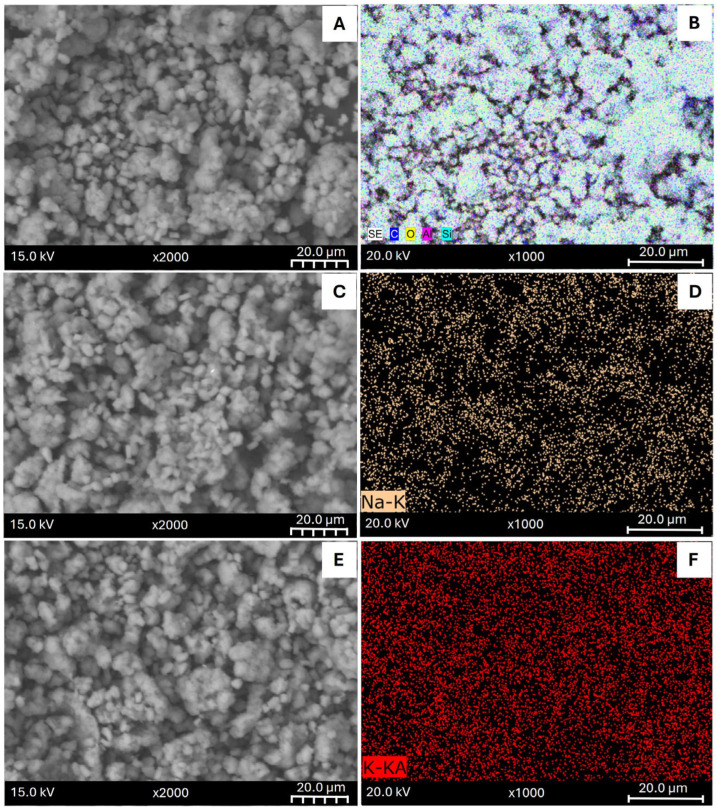
SEM Images (**left**) and Elemental Mapping (**right**) of parent synthetic zeolite (**A**,**B**) and modified samples: SZ_Na (**C**,**D**), SZ_K (**E**,**F**), SZ_Cu (**G**,**H**) and SZ_Zn (**I**,**J**).

**Figure 2 materials-18-02403-f002:**
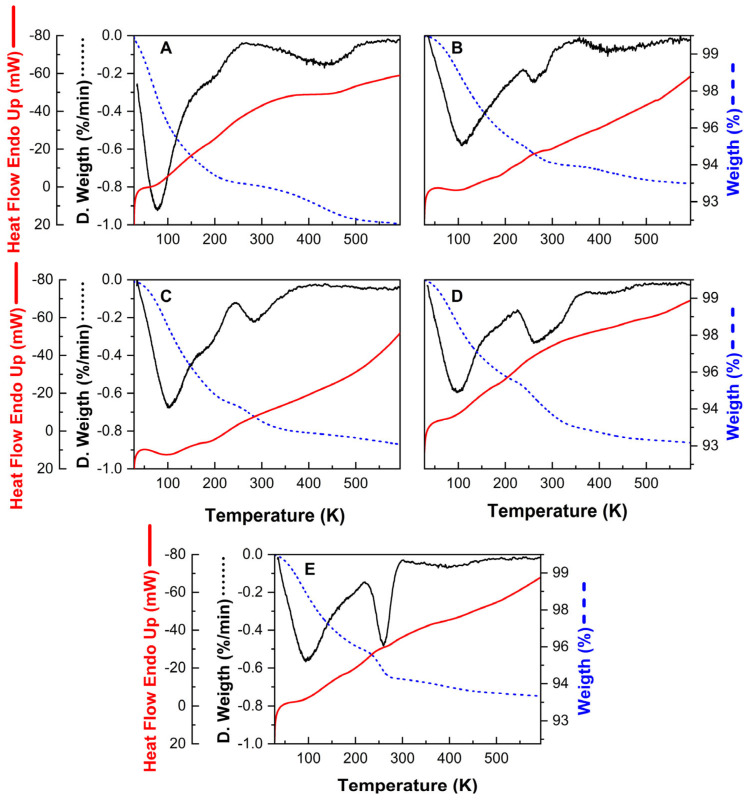
Thermogravimetric Analysis of the parent synthetic zeolite (**A**) and modified samples: SZ_Na (**B**), SZ_K (**C**), SZ_Cu (**D**) and SZ_Zn (**E**).

**Figure 3 materials-18-02403-f003:**
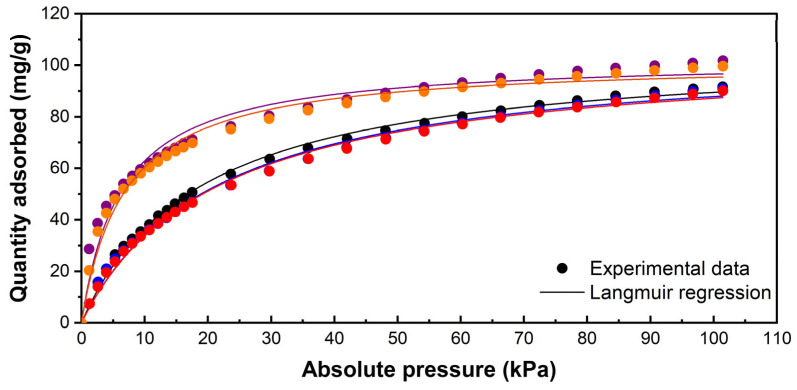
Adsorption isotherms (dots) and Langmuir model representation (lines) of the parent synthetic zeolite (•) and modified samples: SZ_Na (•), SZ_K (•), SZ_Cu (•) and SZ_Zn (•).

**Figure 4 materials-18-02403-f004:**
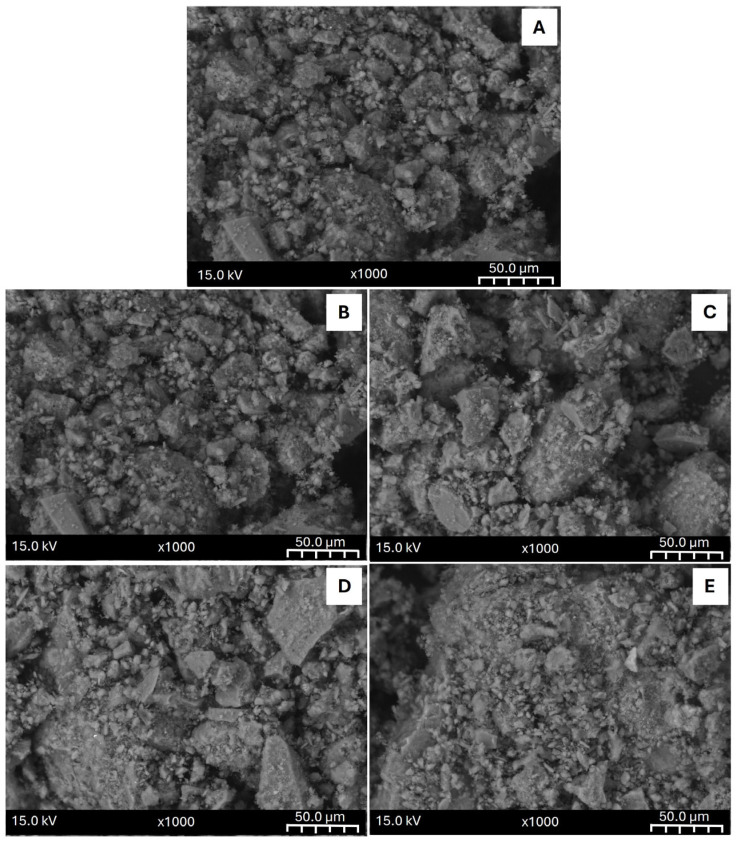
SEM images of parent natural zeolite (**A**) and modified samples: NZ_5 (**B**), NZ_3 (**C**), NZ_8 (**D**) and NZ_4 (**E**).

**Figure 5 materials-18-02403-f005:**
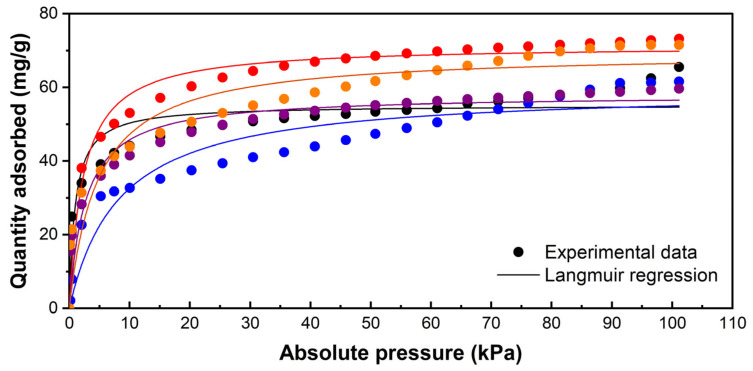
Adsorption isotherms (dots) and Langmuir model representation (lines) of the parent natural zeolite NZ (•) and modified samples: NZ_5 (•), NZ_3 (•), NZ8 (•) and NZ_4 (•).

**Figure 6 materials-18-02403-f006:**
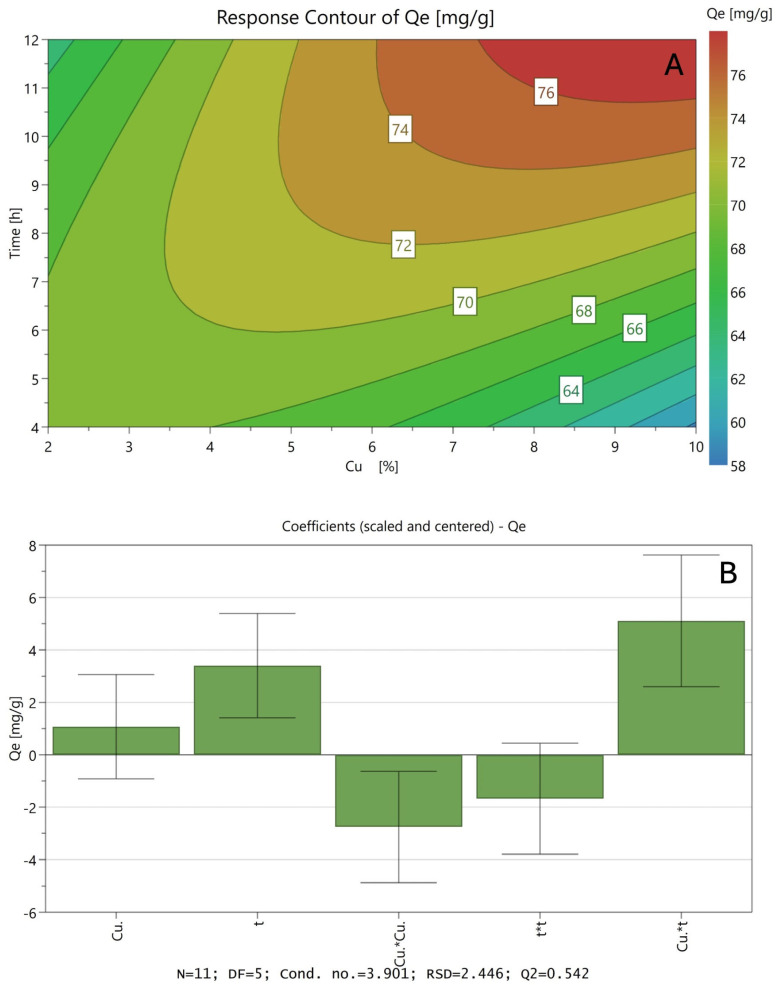
Results of the SRM using a CCD experimental design for CO_2_ adsorption onto natural modified zeolites: Surface Response for the adsorption uptake (**A**) and model coefficient values (**B**).

**Table 1 materials-18-02403-t001:** Synthetic zeolites modified by ion exchange procedure.

Zeolite Sample	Compensation Cation
SZ	NH_4_^+^ *
SZ_Na	Na^+^
SZ_K	K^+^
SZ_Cu	Cu^2+^
SZ_Zn	Zn^2+^

* ZSM-5 zeolites were synthetized by the supplier with NH_4_^+^ as compensation cation.

**Table 2 materials-18-02403-t002:** General and linearized equations of theoretical adsorption models.

Isotherm	General Equation	Linearised Equation
Langmuir	$q_{e}=\frac{Q_{max}K_{L}C_{e}}{1+K_{L}C_{e}}$	$\frac{C_{e}}{q_{e}}=\frac{1}{K_{L}}+\frac{1}{n}$
Freundlich	$q_{e}=k_{F}C_{e}^{1/n}$	$lnq_{e}={lnk}_{F}+\frac{1}{n}lnC_{e}$

**Table 3 materials-18-02403-t003:** Experimental design for natural zeolite modification.

Zeolite Sample	Theoretical Copper Load (%)	Ion Exchange Time (h)	Maximum Adsorption Capacity (mg/g)
NZ_1	2	4	-
NZ_2	10	4	-
NZ_3	2	12	-
NZ_4	10	12	-
NZ_5	0.34	8	-
NZ_6	11.7	8	-
NZ_7	6	2.34	-
NZ_8	6	13.7	-
NZ_9	6	8	-
NZ_10	6	8	-
NZ_11	6	8	-

**Table 4 materials-18-02403-t004:** Physical-chemical and surface properties of parent and modified synthetic zeolites.

Zeolite Sample		SZ	SZ_Na	SZ_K	SZ_Cu	SZ_Zn
^1^ Chemical composition	Si [%]	38.04	40.76	40.10	35.23	37.03
	Al [%]	2.47	2.46	2.63	2.25	2.29
	O [%]	59.49	55.61	55.24	61.32	59.07
	Na [%]	-	1.17	-	-	-
	K [%]	-	-	2.03	-	-
	Cu [%]	-	-	-	1.20	-
	Zn [%]	-	-	-	-	1.61
	Si/Al	15.4	16.6	15.3	15.7	16.2
^2.1^ S_total_ (m^2^/g)		328.3	273.6	291.7	310.6	316.7
^2.2^ A_micro_ (m^2^/g)		214.7	185.5	205.5	213.8	210.9
A_meso_ (m^2^/g)		113.6	88.1	86.2	96.8	105.8
^2.3^ V_total_ (cm^3^/g)		0.181	0.154	0.155	0.169	0.173
^2.4^ V_micro_ (cm^3^/g)		0.163	0.135	0.150	0.156	0.158
V_meso_ (cm^3^/g)		0.018	0.019	0.005	0.013	0.015
^2.5^ Average pore size (nm)		3.263	3.23	3.366	3.333	3.257

^1^ Determined by SEM/EDX; ^2.1^ BET method; ^2.2^ t-plot method; ^2.3^ determined at p/p_0_ = 0.99; ^2.4^ HK method; ^2.5^ BJH method.

**Table 5 materials-18-02403-t005:** Models for CO_2_ adsorption isotherms at 293K on ZSM-5 zeolite, raw and modified.

Model Parameters			SZ	SZ_Na	SZ_K	SZ_Cu	SZ_Zn
Langmuir Model	Q_max_	mg/g	105.8	105.4	105.9	105.9	105.8
	K_l_	dm^3^/mg	0.054	0.049	0.049	0.133	0.128
	R^2^		0.998	0.994	0.996	0.997	0.998
Freundlich Model	n		2.008	2.001	1.962	3.779	3.373
	K_f_	mg/g	10.70	10.21	9.76	31.78	27.77
	R^2^		0.955	0.968	0.967	0.977	0.937

**Table 6 materials-18-02403-t006:** Physical-chemical and surface properties of parent and modified natural zeolites.

Zeolite Sample		NZ	NZ_5	NZ_3	NZ_8	NZ_4
Chemical composition	Si [%]	31.54	33.35	32.36	33.46	34.62
	Al [%]	8.33	9.18	8.64	8.86	8.78
	O [%]	47.6	48.77	50.56	47.83	46.79
	Na [%]	1.74	1.26	1.06	-	0.96
	Mg [%]	0.98	0.78	0.67	0.57	0.59
	K [%]	1.87	1.20	1.51	1.63	1.5
	Ca [%]	3.69	3.44	2.36	2.11	2.08
	Fe [%]	3.99	1.43	1.4	3.35	0.92
	Cu [%]	-	0.59	1.44	2.19	2.76
	Si/Al	3.78	3.63	3.74	3.77	3.94
S_total_ (m^2^/g)		108.3	137.4	146.6	147.4	131.8
V_total_ (cm^3^/g)		0.098	0.119	0.125	0.128	0.114

**Table 7 materials-18-02403-t007:** Models for CO_2_ adsorption isotherms at 293 K on natural zeolite, raw and modified.

Model Parameters			NZ	NZ5	NZ3	NZ8	NZ4
Langmuir Model	Q_max_	mg/g	61.72	60.24	64.93	68.02	73.5
	K_l_	dm^3^/mg	0.232	0.118	0.092	0.164	0.320
	R^2^		0.989	0.977	0.958	0.998	0.990
Freundlich Model	n		6.172	4.849	2.425	4.614	4.48
	K_f_	mg/g	28.96	24.31	10.15	25.86	28.48
	R^2^		0.985	0.983	0.982	0.977	0.999

**Table 8 materials-18-02403-t008:** Results of the experimental design for natural zeolite modification.

Zeolite Sample	Theoretical Copper Load (%)	Ion Exchange Time (h)	Maximum Adsorption Capacity (mg/g)
NZ_1	2	4	70.3
NZ_2	10	4	56.3
NZ_3	2	12	64.9
NZ_4	10	12	73.5
NZ_5	0.34	8	60.2
NZ_6	11.7	8	69.1
NZ_7	6	2.34	62.3
NZ_8	6	13.7	68.0
NZ_9	6	8	72.1
NZ_10	6	8	72.6
NZ_11	6	8	72.3

**Table 9 materials-18-02403-t009:** CO_2_ maximum adsorption uptakes per exposed surface area.

Zeolite Sample	Surface Area (m^2^/g)	Maximum Adsorption Capacity (mg/g)	Maximum Adsorption Capacity (mg/m^2^)
SZ	328.3	105.8	0.32
SZ_Cu	310.6	105.9	0.34
NZ	108.3	61.72	0.56
NZ_4	131.8	73.5	0.56

## Data Availability

The original contributions presented in the study are included in the article, further inquiries can be directed to the corresponding author.

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
