# Peer review of "Modification of Natural and Synthetic Zeolites for CO2 Capture: Unrevealing the Role of the Compensation Cations"

_materials, 2025, doi:10.3390/ma18102403_

Round 1
Reviewer 1 Report
Comments and Suggestions for Authors
The development of effective natural-based adsorbents to address rising CO2 emissions is a significant concern. To improve CO2 capture, synthetic and natural zeolites were modified through ion exchange with various metal cations. Characterization techniques like SEM, EDS, TGA, and Nitrogen Adsorption confirmed the successful incorporation of new metals. CO2 adsorption tests revealed that zeolites modified with bivalent cations (Zn²⁺ and Cu²⁺) outperformed those with monovalent cations (Na⁺ and K⁺). This improved performance is attributed to stronger cation interactions and enhanced field gradients from divalent cations. Optimal adsorption conditions for natural zeolites were determined using Surface Response Methodology. This is a very interesting paper, here are my comments:
The image quality in Fig. 1 requires improvement, and a quantitative analysis of pore sizes should be included for clarity.
Error bars should be added to Fig. 3 to improve data reliability.
Although the paper employs extensive characterization methods, it lacks deeper analysis, such as surface contact angle characterization and potential mechanism insights. It is recommended to compare the findings with existing ref., such as the study by Tetteh, J., Kubelka, J., Qin, L., & Piri, M. (2024), which investigates the effect of ethylene oxide groups on calcite wettability reversal using experimental and molecular dynamics simulation techniques (Journal of Colloid and Interface Science, 676, 408-416).
To better explain the improved adsorption performance due to stronger cation interactions and enhanced field gradients, molecular dynamics simulations should be included to provide additional insights.
The paper contains several grammatical errors that require revision for improved clarity and flow.
Comments on the Quality of English LanguageThe English could be improved to more clearly express the research.
Author Response
The authors sincerely acknowledge the thorough and constructive review provided by the referees. A revised version of the manuscript has been prepared and uploaded, addressing all the suggestions and comments raised. The original manuscript was modified in “Track Changes” mode for transparency. Additionally, a detailed point-by-point response letter has been submitted for the reviewers' consideration.
Thank you for your attention to our work.
Sincerely,
Norberto J. Abreu et al.

Reviewer 2 Report
Comments and Suggestions for Authors
This manuscript presents a study about both synthetic (ZSM-5) and natural Chilean zeolites for enhanced CO₂ capture through ion exchange processes using various compensating cations (Na⁺, K⁺, Cu²⁺, Zn²⁺). The authors combine experimental techniques such as SEM, EDS, TGA, BET surface analysis, and adsorption isotherm modeling, and apply Response Surface Methodology (RSM) to optimize conditions for natural zeolites.
- Could the authors please better specify the nature of the thermal transitions observed in the TGA and DSC profiles?
In particular:
- Were any of the transitions clearly endothermic or exothermic, as revealed by the DSC signal?
- Are the observed mass losses purely related to desorption (e.g., of water or nitrates), or do they involve structural or phase transitions (e.g., glass transition or crystallization phenomena)?
- Have the authors considered whether any of the features in the TGA/DSC curves may be associated with a glass transition (Tg), especially in the case of natural zeolites, which may contain amorphous or poorly crystalline components?
- The novelty of the work should be highlighted better .
- The scale bars in Figure 4 are not visbile.
- Given that the surface area of natural zeolites is lower, but CO₂ adsorption per unit area is higher, could the authors discuss whether the nature or distribution of adsorption sites (rather than total surface) is more relevant in this case?
Comments on the Quality of English Language
moderate revision
Author Response

(The authors gave the same response as above.)

Reviewer 3 Report
Comments and Suggestions for Authors
The manuscript titled “Modification of Natural and Synthetic Zeolites for CO2 Capture: Unrevealing the Role of the Compensation Cations” by Abreu, N.J.; et al. is a scientific work where the authors assessed the carbon dioxide uptake by zeolites and when an ion exchange process was taken place. The morphology, chemistry, thermal stability, adsorption capability were characterized of the raw and modified zeolites with different positive cations. For it, many complementary techniques were devoted in this research. The manuscript is generally well-written and this is a topic of growing interest.
However, it exists some points that need to be addressed (please, see them below detailed point-by-point) to improve the scientific quality of the submitted manuscript paper before this article will be consider for its publication in Materials.
1) Introduction. “The increasing demands of human activities (…) high production rates of carbon dioxide (CO2) and other greenhouses gases” (lines 40-43). Could the authors provide quantitative data insights according to the worldwide economic impact of effects of GHGs with special focus on carbon dioxide? This will significantly aid the potential readers to better understand the signficance of this research.
2) “Among the available solid adsorbents, zeolites and activated carbons stand out due to their exceptional properties (…) Zeolites offer a promising pathway toward suistainable industrial practices” (lines 60-82). Here, I agree with these statements provided by the authors. It should be interesting to discuss how the nanomechanical performance [1] of the zeolites can increase the resistance of concrete for building applications [2]. This will expand the knowledge of this work and strengthen the impact of the gathered results by the authors in this work.
[1] https://doi.org/10.3390/nano13060963
[2] https://doi.org/10.3390/ma17112671
3) Materials & Methods. “Synthetic ZSM-5 zeolite (…) was utilised” (lines 93-95). Why did the authors interrogate the performance of synthetic zeolites instead of the natural-sourced (specially with all the advantages offered by natural zeolites as previously indicated in the line 83)? Some insights need to be provided in this regard.
4) Results & Discussion. “3.1.1. Physical, chemical and surface characterization of raw and modified synthetic zeolites” (lines 196-233). The EDX spectra should be also depicted to better visualize the peaks to quantify the elemantar composition of the raw and modified zeolite samples.
5) Figure 1 (line 226). The lateral scale bar should be enlarge in the SEM micrographs. It is slightly blurry for the potential readers. Same comment for the Fig. 4 (line 342).
6) Figure 3 (line 269). The standard deviation (SD) bars should be added for each tested conditions. Same comment for the Fig. 5 (line 361).
7) Why did the authors not address the compesation of calcium cations (Ca2+) in the zeolites? Some information needs to be detailed in this regard.
8) Finally, did the authors ascertain the selectivity of the modief zeolites to other hazardous atmospheric gases?
9) “5. Conclusions” (lines 432-454). This section perfectly remarks the most relevant outcomes found by the authors in this work. It may be advisable to add a brief statement to remark the potential future action lines to pursue the topic covered in this research and also the promising future prospectives.
Author Response

(The authors gave the same response as above.)

Reviewer 4 Report
Comments and Suggestions for Authors
The paper titled “Modification of Natural and Synthetic Zeolites for CO2 Capture: Unrevealing the Role of the Compensation Cations” is relevant to scientific research, but some modifications are needed to improve the manuscript.
Abstract
- It is necessary to provide more details on the methodology, number of specimens, how they were performed, etc.
- The relationship between the experimental findings and their relevance should be made clearer.
- The statistical analysis was not reported.
Introduction
- I suggest reinforcing the importance of zeolite modification and its practical impact.
- It is important to present more clearly the relationship between CO₂ emissions, the limitations of existing technologies and the relevance of zeolites.
- Highlight how the research aligns with the Sustainable Development Goals (SDGs).
- I suggest that the authors clearly describe the limitations of previous studies and justify the need for this study.
- The null hypothesis is not present.
Methods
- How was the sample calculation performed? Was the methodology based on a previously published study?
- It is necessary that the experimental parameters chosen (time, temperature, concentration) be justified.
- A clearer explanation of the application of RSM and CCD is recommended.
- Detail the precision of the equipment used.
- I suggest that the authors clarify the choice of the Langmuir and Freundlich isothermal models.
- It is important to compare the experimental parameters with previous studies.
- Highlight the relevance of the parameters studied in the modification of zeolites.
- The statistical analysis was not reported.
Results and Discussion
- I recommend that authors review the submission guidelines of this journal and rewrite the results and discussion sections separately, so that it is possible to review these sections.
Conclusion
- I suggest that the authors rewrite the conclusion in a clear and direct manner, based on the results obtained and conveying the authors' opinion on the data found in the study.
Author Response

(The authors gave the same response as above.)

Round 2
Reviewer 1 Report
Comments and Suggestions for Authors
The revised manuscript appears to include only minor textual edits, without addressing the core issues raised in the previous review. There is no evidence of substantial improvement in the scientific content or overall structure of the paper. In particular, the SEM images presented in Figure 1 remain poorly focused and lack sufficient clarity to support the claims made in the manuscript. Additionally, the organization and logical flow of the paper have not been significantly enhanced. Given the limited effort put into the revision and the persisting concerns regarding data quality and presentation, I do not recommend acceptance at this stage.
Author Response
We appreciate the reviewer’s feedback, and the time dedicated to evaluating our revised manuscript. We are concerned that the improvements made were not perceived as sufficiently addressing the concerns raised. A point-by-point answer is developed in the attached document.
Thank you for your attention to our work.
Sincerely,
Norberto J. Abreu et al.

Reviewer 2 Report
Comments and Suggestions for Authors
I recommend the publication of this article
Author Response
The authors sincerely acknowledge the thorough and constructive review provided by the referees. Thank you for your attention to our work.
Sincerely,
Norberto J. Abreu et al.
Reviewer 4 Report
Comments and Suggestions for Authors
After review, with all questions answered and suggestions accepted, I recommend publishing the article.
Author Response

(The authors gave the same response as above.)
